# Harnessing the Power of Enteric Glial Cells’ Plasticity and Multipotency for Advancing Regenerative Medicine

**DOI:** 10.3390/ijms241512475

**Published:** 2023-08-05

**Authors:** Marie A. Lefèvre, Rodolphe Soret, Nicolas Pilon

**Affiliations:** 1Département des Sciences Biologiques, Université du Québec à Montréal (UQAM), Montreal, QC H3C 3P8, Canada; lefevre.marie@courrier.uqam.ca; 2Centre D’excellence en Recherche Sur Les Maladies Orphelines—Fondation Courtois (CERMO-FC), Université du Québec à Montréal, Montreal, QC H2X 3Y7, Canada; 3Département de Pédiatrie, Université de Montréal, Montreal, QC H3T 1C5, Canada

**Keywords:** neural crest cells, Schwann cells, enteric glial cells, diversity, plasticity, multipotency, regenerative medicine, Hirschsprung disease, inflammatory bowel diseases

## Abstract

The enteric nervous system (ENS), known as the intrinsic nervous system of the gastrointestinal tract, is composed of a diverse array of neuronal and glial cell subtypes. Fascinating questions surrounding the generation of cellular diversity in the ENS have captivated ENS biologists for a considerable time, particularly with recent advancements in cell type-specific transcriptomics at both population and single-cell levels. However, the current focus of research in this field is predominantly restricted to the study of enteric neuron subtypes, while the investigation of enteric glia subtypes significantly lags behind. Despite this, enteric glial cells (EGCs) are increasingly recognized as equally important regulators of numerous bowel functions. Moreover, a subset of postnatal EGCs exhibits remarkable plasticity and multipotency, distinguishing them as critical entities in the context of advancing regenerative medicine. In this review, we aim to provide an updated overview of the current knowledge on this subject, while also identifying key questions that necessitate future exploration.

## 1. Introduction

The enteric nervous system (**ENS**) is the most complex division of the peripheral nervous system [1,2], being composed of thousands of interconnected ganglia that contain variable numbers of neurons and glia. In mammals, these ganglia are organized into two interconnected networks known as the myenteric plexus and the submucosal plexus. The myenteric plexus is located between the longitudinal and circular muscle layers of the gastrointestinal tract, and is thereby responsible for regulating mixing and peristaltic patterns of contraction and relaxation. From its position under the mucosa, the submucosal plexus instead regulates various epithelial functions, including selective permeability [2,3]. It is also noteworthy that the ENS interacts bilaterally with both the immune system and the microbiota [4,5,6,7,8,9]. Moreover, while the ENS largely operates independently of the central nervous system, it plays a crucial role in facilitating bidirectional gut-brain communication via extrinsic afferent and efferent nerves of the vagal, spinal/sympathetic and pelvic pathways [10,11].

The diverse range of gastrointestinal functions controlled by the ENS can be partially attributed to the similarly wide variety of enteric neuron subtypes, with more than 20 functional classes identified in mammals [12,13,14]. For instance, smooth muscle contraction and relaxation are mainly regulated by cholinergic (excitatory) and nitrergic (inhibitory) neuron subtypes, respectively. However, it is also necessary to acknowledge the importance of enteric glial cells (**EGCs**) in the regulation of several bowel functions [15,16,17]. A growing body of evidence even suggests that some of the key functions attributed to the ENS are in fact primarily performed by EGCs. In particular, recent studies convincingly demonstrate that EGCs can directly control gastrointestinal immunity [18] and epithelial integrity [19], notably by sensing insults/damages and subsequently releasing relevant pro-resolution diffusible factors [9,20,21,22].

Currently, up to nine subtypes of EGCs can be distinguished in the mature mammalian ENS based on morphological/topological (Table 1 and Figure 1), functional (Table 2) or transcriptional (Table 3) criteria. As for enteric neuron subtypes, such diversity strongly suggests that EGCs regulate specific gastrointestinal functions in a subtype-dependent manner, but this question is only beginning to be addressed. As cell-based therapies are increasingly being considered for treating various gastrointestinal conditions, it has also become mandatory to begin studying the mechanisms of EGC formation and diversification in the mammalian ENS. This is especially important when considering the plasticity and multipotency exhibited by some EGCs. In this review, we summarize the most relevant findings along all these lines and identify what we think are the most urgent needs for future research.

## 2. The ENS Is Built from Multiple Progenitors

Based on several studies using various model organisms, there is a large consensus that most of the ENS is derived from neural crest cells (**NCCs**)—a vertebrate-specific population of multipotent embryonic cells named in reference to their initial accumulation in the dorsal midline of the developing neural tube. According to their location along the anterior–posterior axis, NCCs can be subdivided into four main subpopulations (from anterior to posterior): cranial, vagal, trunk and sacral. Each of these four subpopulations have distinctive differentiation potential and migratory pattern from the dorsal neural tube [32,33]. Three of them (vagal, trunk and sacral) are a source of ENS progenitors, with vagal NCCs being responsible for generating the majority of enteric neurons and glia [33,34,35,36,37,38]. In mice, vagal NCCs first invade the foregut mesenchyme around embryonic day (e) 9.5 and then migrate posteriorly toward the hindgut to reach the prospective rectum around e14.5 [39]. Sacral NCCs contribute a small subset of enteric neurons and glia in the opposite direction (i.e., posterior to anterior), entering the hindgut mesenchyme around e13.5 and then intermingling with the vagal NCC-derived ENS progenitors at e14.5 [40]. 

Of note, a subset of vagal NCCs also migrate along the mesentery during the e9.5–e14.5 period [41,42,43,44], hence entering the developing gut at multiple sites along the anterior–posterior axis [44]. Multiple points of entry similarly characterize the contribution of subsets of both vagal and trunk NCCs that have adopted a Schwann cell precursor (**SCP**) intermediate state, and thus colonize the developing gut via extrinsic nerve tracts [45,46]. SCPs of trunk origin appear to be the latest to colonize the developing gut [46], contributing to the ENS only after some of the other ENS progenitors from the myenteric plexus have migrated radially inward to form the submucosal plexus (from e15.5 onwards) [47,48].

Information about potential non-NCC sources of ENS progenitors is scarce. There is one study reporting a contribution from the ventral neural tube in chick embryos [49], while a more recent study reports a contribution from gut/pancreas endodermal epithelium in mouse embryos [50]. In both cases, the contribution to the developing ENS was mentioned to occur slightly after the colonization by vagal NCCs [49,50]. This contribution appeared minor and regionally restricted, either radially for the endoderm source (in the myenteric plexus only) [50] or along the anterior–posterior axis for the ventral neural tube source (in the foregut only) [49]. Moreover, the endoderm source seems further limited in terms of differentiation potential, contributing some enteric neurons but no glia [50].

## 3. Formation and Diversification of EGCs

The differentiation of ENS progenitors into enteric neurons and glia mostly occurs during gut colonization, before birth. For over 20 years, it has been known that this process is strongly skewed toward a neurogenic fate, as notably evidenced by the significant delay between the appearance of the earliest markers of committed enteric neurons and those of EGCs, reaching ~2 days in the foregut of mouse embryos (e10.0–10.5 vs. e11.5–12.0, respectively) [51,52]. This neurogenic bias suggests that early enteric gliogenesis must first involve countermeasures against the pro-neuronal molecular machinery. Yet, we are still virtually blind about how exactly the neurogenic-to-gliogenic fate transition is molecularly orchestrated in the developing ENS, with very little advancement over the past years [53]. 

FABP7 (Fatty Acid Binding Protein 7) is generally considered as the earliest marker of committed EGCs [51]. However, this metabolic protein is most likely not playing an active role in the regulation of EGC differentiation, per se. Such a role must instead be imparted to a gene regulatory network involving specific transcription factors and signaling pathways. Efforts to assemble a gene regulatory network for vagal NCCs and their derived ENS progenitors have begun [54], but we are very far from this level of precision in the case of enteric gliogenesis. Indeed, as reviewed a few times over the past years [53,55,56], only a handful of relevant positive regulators are currently known and/or suspected to play a role in enteric gliogenesis (Table 4). Moreover, as again recently evidenced by spatially restricted transcriptome studies [57], many of these pro-glial regulators are also expressed in ENS progenitors (and/or in NCCs/SCPs before they colonize the developing gut), thereby complicating their functional analysis in committed EGCs. In the context of constitutive loss-of-function experiments, this is notably reflected by early defects in ENS progenitors that preclude and/or confound the analysis of enteric gliogenesis. This problem of reiterated roles can in theory be addressed by Cre/LoxP-based conditional loss-of-function approaches, but this possibility remains somehow limited for the same reason; there is currently no Cre driver line with exclusive activity in EGCs and not in ENS progenitors. Alternatively, gain-of-function experiments appear to be a relatively simple way for acquiring information about enteric gliogenesis, as notably demonstrated for the NR2F1 transcription factor [58] and the Hedgehog signaling pathway [59,60]. 

Hopefully, single-cell transcriptomics-based analyses of the developing mammalian ENS [90,91,92,93] will help to identify new candidate regulators of the neurogenic-to-gliogenic fate transition. These findings should pave the way for future functional investigations, particularly as more comprehensive and glia-focused studies across multiple stages are starting to emerge [94,95]. This will most likely take longer to identify candidate regulators of EGC diversification, notably because we do not yet really know when the different subtypes of EGCs begin to appear. Another difficulty is that EGC diversification can take different forms in mammals (i.e., morphological/topological, functional or transcriptional; see Table 1, Table 2 and Table 3), which are currently hard to reconcile. Perhaps a meta-analysis of all transcriptional subtypes that are listed in Table 3 might be helpful, but the high heterogeneity of experimental conditions that were used to generate these data makes this quite unlikely. Moreover, there is a need to systematically validate these transcriptomic data by immunofluorescence analyses aimed at visualizing the spatiotemporal distribution of the associated proteins. Notably, this would allow to address the question of whether the different topo-morphological EGC subtypes could also be identified with specific markers. A pilot experiment shows that this is not the case for SLC18A2, which has previously allowed to define a transcriptional subtype of EGCs [13] with likely neurogenic potential [30] but that we have found not to be confined to a particular topo-morphological subtype (Figure 2).

Now that we are more aware of the diverse sources of ENS progenitors, we should also consider the possibility that these different origins might contribute to EGC diversification as well. Although clonal analysis of ENS formation in the small intestine suggests that all four topo-morphological subtypes of EGCs can be engendered by a single ENS progenitor [92], this does not mean that all types of ENS progenitors contribute equally in all segments of the gastrointestinal tract. Moreover, it is important not to forget the additional potential contribution of structural and/or environmental changes that occur within the gut wall and/or the lumen during gut morphogenesis (at both prenatal and postnatal stages). A good example is the impact of the microbiota, which was shown to be key for attracting EGCs in the mucosa (topo-morphological EGC Type III_(Mucosa)_) of mice around weaning age [96]—although this specific mode of regulation is not universal, being notably absent in humans [97].

## 4. Plasticity and Multipotency of EGCs

### 4.1. EGCs’ Plasticity

Once generated and integrated in the mature ENS, EGCs are not static. On the contrary, EGCs exhibit a high level of phenotypic plasticity, which we here define by changes in molecular composition, structure and/or function. Under physiological conditions, EGCs’ plasticity is not obvious at first glance, with a single study reporting dynamic GFAP expression in murine topo-morphological Type 1 EGCs [23]. As recently reviewed in more detail elsewhere [20,22], the plasticity of EGCs is instead primarily evidenced under pathological circumstances, such as intestinal inflammation or infection, which trigger reactive gliosis. In addition to transient changes in the expression of glial markers (e.g., GFAP, S100β) [98], reactive EGCs can be characterized by changes in morphology (e.g., increased length and thickness of glial processes) [99], secretion of pro-inflammatory mediators (e.g., IL-1B, IL-6, NO) [100,101,102], immune competence (e.g., T lymphocyte activation via surface expression of MHC-II) [103], proliferative activity [104] or pro-apoptotic potential [105].

Depending on context, these changes are believed to have either detrimental effects by exacerbating pathological inflammatory processes or beneficial effects by neutralizing inflammation and promoting repair [22]. Accordingly, as indicated in Section 5, some of these aspects of reactive enteric gliosis are currently considered potential therapeutic targets for several gastrointestinal diseases. However, optimizing such approaches will require a better understanding of how the different EGC subtypes respond to gliosis triggers. As such responses are likely variable as a function of EGC subtypes, this knowledge might pave the way to more precise interventions restricted to single EGC subtypes.

### 4.2. EGCs’ Multipotency

In addition to their extensive phenotypic plasticity, a subset of EGCs have the remarkable capacity to self-renew and differentiate into enteric neurons. Current knowledge suggests that this subset of EGCs with stem cell-like properties corresponds to what was initially reported to be a population of postnatal/adult ENS stem cells in mice [106,107,108] and humans [109,110,111]. As outlined in Table 5, the stem cell-like properties of EGCs vary as a function of experimental conditions in mice, being virtually undetectable under steady-state conditions in vivo. Yet, proliferation and neuronal differentiation of adult EGCs do exist during homeostasis in zebrafish [112], suggesting that these properties were somehow attenuated during vertebrate evolution. The stem cell-like properties of mammalian EGCs are nonetheless especially obvious in vitro, where EGCs sorted from adult bowels can not only be differentiated into neurons and glia but also into myofibroblasts [113]—as also noted in the early reports of postnatal/adult ENS stem cells [106,107,108,109]. Whether postnatal EGCs have this capacity to generate myofibroblasts in vivo is currently unknown. If it exists, this differentiation potential will probably require special circumstances to be revealed. Smooth muscle injury would most likely be a prerequisite in this case, just like ENS injury appears required to awake the self-renewing and neurogenic potential of EGCs in mice [114,115,116].

Further research is clearly necessary to fully understand both the nature and the regulatory mechanisms of EGCs’ stem cell-like properties in mammals. One especially important question to address is whether the self-renewal and multipotency of EGCs seen at the population level are combined in a specific EGC subtype or are instead divided in different EGC subtypes. Comparison of thymidine analog incorporation assays and cell lineage tracing studies suggest that neuronal differentiation from EGCs mostly occurs independently of cell proliferation [113,114,116], but both types of analyses will need to be combined to clearly establish the extent of such trans-differentiation capacity. In connection with this, are EGC-derived neurons exclusively made from the neurogenic EGC subtypes recently identified by scRNA-seq [30]? Do each of the two neurogenic EGC subtypes identified in this study generate mutually exclusive neuron subtypes? Similar questions specifically arise for the self-renewal of EGCs. Is it an intrinsic property of all topo-morphological subtypes of EGCs? Responses to all these questions will be required to take full advantage of EGCs’ multipotency for therapeutic purposes.

## 5. Taking Advantage of EGCs’ Plasticity and Multipotency for Therapeutic Purposes

### 5.1. Control of Inflammation and Infection in the Gastrointestinal Tract

As mentioned in the previous section, reactive EGCs are involved in the pathogenesis of various gastrointestinal disorders [22]. In the case of IBD (inflammatory bowel diseases, which include ulcerative colitis and Crohn’s disease), reactive EGCs primarily adopt a pro-inflammatory phenotype that exacerbates both innate and adaptative immune responses [21]. Similar observations have also been made in the context of IBS (irritable bowel syndrome) [118,119] and POI (postoperative ileus) [120]. Although there are no specific drugs/products that specifically target EGCs, several studies have nonetheless successfully modulated the detrimental effects of enteric gliosis for therapeutic purposes [22,121]. For example, Pentamidine, a broad-spectrum anti-infective small molecule that targets and inhibits S100β, can prevent 5-Fluorouracil-induced intestinal mucositis and associated enteric neurotoxicity by decreasing S100β secretion from reactive EGCs, thereby attenuating downstream RAGE/NF-κB signaling [122]. Interestingly, not only conventional drugs but also nutraceutical products have shown promising effects in modulating the pathological effects of reactive EGCs [121]. For instance, the cannabinoid-related PEA (palmitoylethanolamide, found in soybeans and peanuts) was reported to exert an anti-inflammatory effect in the context of ulcerative colitis by targeting and activating PPARα which then inhibits S100β production/secretion from reactive EGCs [123]. Of note, PEA also proved useful in the case of HIV-1 Tat-induced diarrhea via the same PPARα-dependent mechanism in reactive EGCs [124]. 

While most therapeutic strategies in this area focus on mitigating the deleterious effects of reactive EGCs, it should not be forgotten that these cells may also have beneficial effects that might be taken advantage of. One especially appealing possibility would be to control the secretion of GDNF, which was found to be turned on in reactive EGCs in the context of Crohn’s disease [125], and whose beneficial effects on restoring epithelial barrier integrity in this same pathological context are well known [126,127]. 

### 5.2. Repair and Regeneration of the ENS

The discovery of postnatal/adult ENS stem cells [106,107,108,109,110,111] has sparked great interest for the development of cell transplantation-based therapies aimed at regenerating the damaged/missing ENS. We now assume that this stem/progenitor cell population is mostly composed of intrinsic EGCs (Table 5), but at least a minor contribution from extrinsic Schwann cells is also likely. Indeed, extrinsic Schwann cells are closely associated with intestinal tissues and are often labeled with the same transgenic markers (driven by *Plp1*, *Nestin* or *Sox10* regulatory sequences) used to label EGCs, and thus are hard to be excluded from gastrointestinal cell preparations. Moreover, reminiscent of the normal capacity of SCPs to form enteric neurons during late ENS development [46], Schwann cells from adult peripheral nerves can be grown as neurospheres and differentiated into neurons both in culture and when transplanted in the mouse gastrointestinal tract in vivo [128].

Mouse models of Hirschsprung disease have been the preferred tools for testing and developing cell transplantation-based therapies [128,129,130,131,132,133,134], although diseases with less severe phenotypes (e.g., oesophageal achalasia, gastroparesis) are now increasingly recognized as likely being more amenable to therapy in a real-world setting [135]. Hirschsprung disease is characterized by the complete lack of ENS ganglia over varying lengths of the rectum and distal colon, due to incomplete colonization by vagal NCC-derived ENS progenitors [33,136]. Yet, the so-called aganglionic segment is naturally enriched in Schwann cells owing to the overabundance of extrinsic nerves in this context [137]. This has important practical implications for highly desirable autologous cell-based therapies, explaining why not only the ENS-containing region [132], but also the ENS-devoid region [138,139], can be a source of ENS stem/progenitor cells likely enriched in EGCs and Schwann cells, respectively. However, it is currently unclear if both sources can generate the same complement of enteric neuron subtypes after ex vivo expansion and in vivo transplantation. SCP-derived enteric neurons are normally strongly biased towards an excitatory CALR+ phenotype, with only minimal contribution to the inhibitory NOS1+ pool [46]. Although cell culture can reprogram the cell differentiation potential, the extent of derivatives made from EGC- and Schwann cell-derived ENS stem/progenitor cells might nonetheless remain skewed somehow. The same question also applies to the diversity of EGC subtypes that can be engendered from each source of ENS stem/progenitor cells. 

One possibility for maximizing neuronal and glial diversification—and, hence, functional recovery of the reconstituted ENS—would be to co-transplant ENS stem/progenitor cells of different origins, as recently experimented for vagal and sacral NCC-derived ENS progenitors differentiated from human pluripotent stem cells [134]. That being said, in situ stimulation of tissue-resident ENS stem/progenitor cells appears as a much simpler approach to address this issue, and GDNF proved to be a potent trigger in this context [140,141]. Indeed, rectal administration of GDNF over a relatively short period of time after birth (five days) induced a new functional ENS in the otherwise aganglionic colon of three genetically distinct mouse models of Hirschsprung disease (*Piebald-Lethal* [142], *Holstein* [143] and *TashT* [144]). This treatment stimulated neurogenesis and gliogenesis in both aganglionic and hypoganglionic segments [140,141], generating several neuronal subtypes in the aganglionic zone while also correcting the cholinergic vs. nitrergic neuronal imbalance normally found in the upstream hypoganglionic zone [141,145]. Intriguingly, genetic cell lineage tracing studies using the Schwann cell-specific Cre driver *Dhh-Cre* revealed that only about a third of GDNF-induced neurons are derived from this lineage in the aganglionic segment. Moreover, combined EdU incorporation assays showed that the majority of GDNF-induced neurons were not derived from a dividing precursor. Other data suggest that sacral NCC-derived EGCs might also be present in the aganglionic segment [141], but their contribution to the regenerative process, if any, is currently unknown. Like for cell transplantation-based therapies, it also remains to be known if Schwann cell- and EGC-derived ENS stem/progenitor cells generate their own set of neuronal and glial subtypes. Addressing these questions in the context of Hirschsprung disease will also be important for improving our general knowledge of ENS stem/progenitor cells.

## 6. Conclusions and Perspectives

EGCs are now recognized to be almost as important as enteric neurons in orchestrating several gastrointestinal functions, but we still know very little about how these functions are taken in charge by the different EGC subtypes that were noted recently. As more and more tools and datasets are being generated, the field seems to have entered a new era which should soon yield significant breakthroughs. Increasing our knowledge of EGC formation and function will be important not only for managing numerous gastrointestinal diseases but also potentially for many neurological disorders involving protein aggregates, like Parkinson disease [146,147,148] or amyotrophic lateral sclerosis [149,150,151], which are suspected to start in the ENS before spreading in the brain via extrinsic nerves—either directly (via retrograde transport of protein aggregates) or indirectly (via gut microbiota-derived metabolites) [152]. For example, since reactive EGCs are likely involved during the earliest stages of both Parkinson disease [153,154] and amyotrophic lateral sclerosis [151] like they are in IBD, the development of therapeutic strategies targeting these cells might hence be useful in all of these contexts.

## Figures and Tables

**Figure 1 ijms-24-12475-f001:**
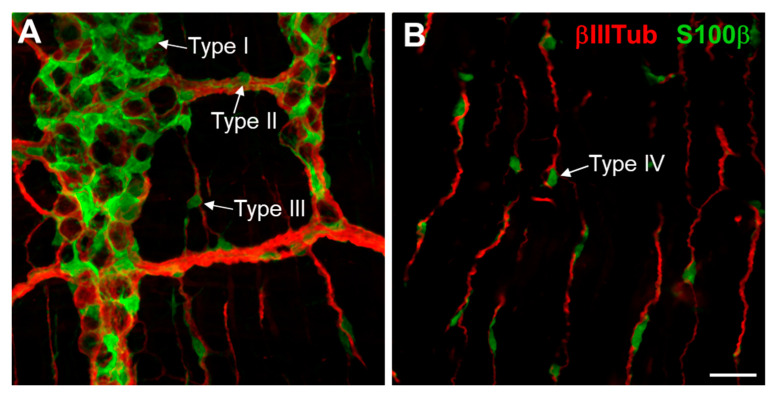
Tissue distribution of the 4 topo-morphological subtypes of EGCs in murine duodenum at P20. (**A**,**B**) Double immunofluorescence staining of S100β-positive EGCs and βIII-Tubulin-positive enteric neurons and nerve fibers at the level of myenteric ganglia (**A**) and circular muscle layer (**B**). As indicated by arrows, EGCs Type 1 are located in myenteric ganglia, Type II in thick interganglionic nerve fibers, while Type III and Type IV are both associated with thin extraganglionic neuronal fibers. Scale bar, 50 µm.

**Figure 2 ijms-24-12475-f002:**
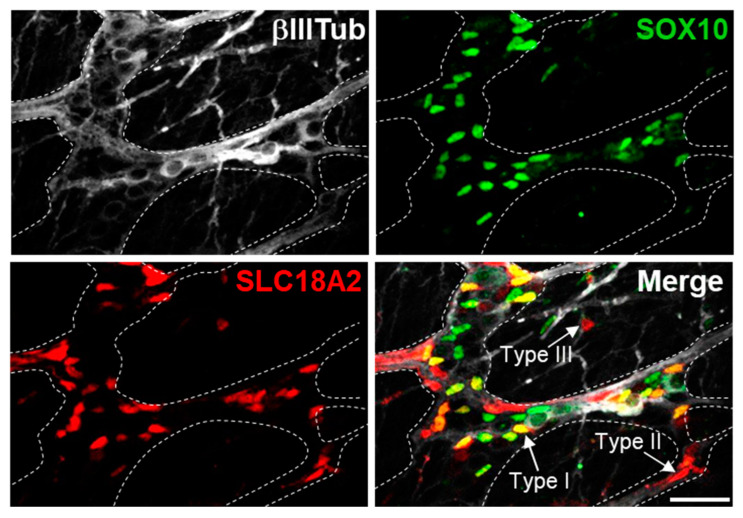
Distribution of SLC18A2 protein in the myenteric plexus of WT mice at P10. Triple immunofluorescence staining shows that SLC18A2 predominantly labels SOX10-positive EGCs Type 1, but also SOX10-negative EGCs Type II and Type III (see arrows in merge panel). βIII-Tubulin labels enteric neurons and nerve fibers. Scale bar, 100 µm.

**Table 1 ijms-24-12475-t001:** Topo-morphological subtypes of EGCs in mice *.

EGC Subtype	Anatomical Region	Topological and Morphological Features
Type I	Myenteric and submucosal plexuses	Within myenteric and submucosal ganglia; composed of multiple irregular and highly branched processes, terminating with end-feet like structures and contacting multiple EGCs and neurons. Also named “protoplasmic”.
Type II	Myenteric and submucosal plexuses	Located within or at the border of interganglionic fibers; exhibiting long parallel processes extending along interganglionic fibers without ensheathing them. Also named “fibrous”.
Type III_(MP/SMP)_	Myenteric and submucosal plexuses	Outside ganglia and interganglionic fibers, but lying in the same plane; displaying four major processes with secondary branching, closely associated with thin neuronal fibers or small blood vessels.
Type III_(Mucosa)_	Lamina propria	
Type IV	Circular and longitudinal smooth muscle layers	Associated with thin nerve fibers in the muscularis; characterized by two unbranched processes extending parallelly along nerve fibers. Also named “bipolar”.

* Based on. [23,24].

**Table 2 ijms-24-12475-t002:** Functional subtypes of EGCs in mice.

References	Characterized Function	Functional Specialization of EGC Subtypes
Boesmans et al., 2015 [23]	Calcium responsiveness to ATP stimulation	In the adult mouse colon, the topo-morphological types I, II and III from myenteric plexus display subtype-specific calcium responsiveness, with Type I EGCs being the most responsive and Type III being the least responsive to purinergic receptor stimulation.
Seguella et al., 2022 [25]	Calcium responsiveness to ADP and CCK stimulation	Type I EGCs exhibit four distinct profiles of calcium responsiveness to ADP and CCK stimulation (ADP^High^/CCK^High^, ADP^High^/CCK^Low^, ADP^Low^/CCK^High^, ADP^Low^/CCK^Low^) in adult mice, this local diversity being also differentially distributed between duodenum and colon (regional diversity).
Baghdadi et al., 2022 [19]	Intestinal epithelium homeostasis and repair	GFAP+ Type III_(Mucosa)_ EGCs are a key component of the intestinal stem cell niche in the adult murine ileum, being a source of WNT signals important for epithelium homeostasis and repair.

Abbreviations: ADP, adenosine diphosphate; ATP, adenosine triphosphate; CCK, cholecystokinin.

**Table 3 ijms-24-12475-t003:** Transcriptional subtypes of EGCs in mice and humans.

References	Experimental Condition	Transcriptional Signatures
Zeisel et al., 2018 [26]	scRNA-seq of tdTomato+ cells from small intestine muscles and myenteric plexus of P21 *Wnt1-Cre;R26R-tdTomato* mice.	7 glial clusters (ENTG1-7), including 1 proliferating and 3 expressing *Slc18a2*.
Drokhlyansky et al., 2020 [13]	snRNA-seq of GFP+ nuclei and ribosome-bound RNA from full-thickness small intestine and colon of *Sox10-Cre;INTACT*, *Wnt1-Cre2;INTACT*, and *Uchl1-H2BmCherry:GFPgpi* mice, as a function of age (11–14 weeks vs. 50–52 weeks), sex and circadian phase.	3 glial clusters (Glia1-3) enriched in *Gfra2*, *Slc18a2*, or *Ntsr1* transcripts, respectively. No difference as a function of gut region, age, sex, or circadian phase.
Wright et al., 2021 [27]	snRNA-seq of mCherry+ nuclei from distal colon muscles and myenteric plexus of P47-52 *Wnt1-Cre;R26R-H2B-mCherry* mice.	4 glial clusters (Glia1-4); no clearly distinctive features reported.
Baghdadi et al., 2022 [19]	Re-analysis of scRNA-seq data generated using colonic mesenchymal cells isolated from the mucosa of adult WT mice [28].	3 glial clusters (EGC#0-EGC#2) based on % of *Gfap*- and *Plp1*-expressing cells in each cluster: Gfap^High^/Plp1^Mid^, Gfap^Low^/Plp1^High^, and Gfap^Mid^/Plp1^Low^.
	Re-analysis of scRNA-seq data generated using colonic stromal cells isolated from mucosal biopsies of healthy humans and patients with ulcerative colitis (UC), aged between 18 and 90 years [29].	4 glial clusters (hEGC#0-EGC#3) based on health status, with hEGC#1 and 2 enriched in healthy samples and hEGC#0 and 3 enriched in UC samples. hEGC#1 corresponds to murine EGC#1 (Gfap^Low^/Plp1^High^), while hEGC#0 corresponds to murine EGC#0 (Gfap^High^/Plp1^Mid^).
Guyer et al., 2023 [30]	scMulti-seq (scRNAseq combined with ATAC-seq) of GFP+ cells from small intestine muscles and myenteric plexus of P14 *Plp1-GFP* mice.	9 transcriptional clusters (clusters #0–8) based on gene expression, chromatin accessibility at neuronal marker peaks, and motif enrichment patterns, including: 2 classified as replicating, 4 with open chromatin, 1 with restricted chromatin and 2 poised for neurogenesis. One of these “neurogenic” clusters is specifically enriched in *Slc18a2*, *Ramp1*, and *Cpe* transcripts.
Schneider et al., 2023 [31]	scRNA-seq of GFP+ cells from full-thickness colon of adult *Sox10-Cre;INTACT* mice kept under restraint stress or not.	4 glial clusters, including 1 exclusively present under psychological stress condition, named enteric glia, associated with psychological stress (eGAPS) and highly expressing *Nr4a1/2/3*.

Abbreviations: ATAC-seq, Transposase-Accessible Chromatin with sequencing; GFP, green fluorescent protein; scRNA-seq, single-cell RNA sequencing; snRNA-seq, single-nucleus RNA sequencing.

**Table 4 ijms-24-12475-t004:** List of previously reported regulators of enteric gliogenesis in mice.

EGC Regulator	Relevant Expression Pattern	Experimental Evidence
ASCL1 (MASH1) Transcription factor	NCCs [61], ENS progenitors [62,63], enteric neurons and EGCs [62].	In addition to defective neurogenesis, *Ascl1^−/−^* embryos have less S100B+ Sox10+ EGCs in ileum and colon. Rescue of enteric neurogenesis but not gliogenesis in *Ascl1^KINgn2^* embryos suggests that ASCL1, which is typically pro-neuronal, also plays an active role in promoting gliogenesis [62].
FOXD3 Transcription factor	NCCs [64,65], SCPs [65], ENS progenitors [66] and EGCs [67].	Targeted deletion of *Foxd3* in vagal NCC-derived ENS progenitors specifically impairs the formation of S100β+ EGCs in *Foxd3^flox/−^;Ednrb-iCre;R26R^YFP+^* embryos, leaving neurogenesis virtually unaffected [67].
NR2F1 Transcription factor	NCCs [68] and SCPs [69].	Insertional mutagenesis-induced upregulation of *Nr2f1* in NCCs leads to premature formation of S100β+ SOX10+ EGCs at the expense of SOX10+ ENS progenitors in *Nr2f1^Spt/Spt^* embryos [58].
SOX10 Transcription factor	NCCs [70,71], SCPs [71], ENS progenitors [70,72] and EGCs [51].	ENS progenitors from *Sox10^LacZ/+^* embryos precociously express the pan-neuronal marker PGP9.5 [72]. Decreased SOX10 levels attenuate the Hedgehog-induced expression of the EGC marker *Fabp7* in *Wnt1-Cre;Sufu^f/f^;Sox10^N/+^* embryos [59].
TBX3 Transcription factor	NCCs [73], ENS progenitors and enteric neurons [27,74,75].	Targeted deletion of *Tbx3* in NCCs leads to a marked reduction of S100β+ EGC density in *Wnt1-Cre;Tbx3^fl/fl^* embryos. Detection of TBX3 protein in enteric neurons but not in EGCs suggest a non-cell autonomous role [74].
Hedgehog Signaling pathway	NCCs [76] and ENS progenitors [77] for PTCH1/SMO binding/signaling receptors and GLI nuclear effectors. Gut epithelium for SHH and IHH ligands [78,79].	*Ptch1* deletion-induced activation of Hedgehog signaling in vagal NCCs upregulates the EGC marker *Fabp7* in the developing gut of *b3-IIIa-Cre;Ptch1^f/f^* embryos, while transduction of CRE in cultured *Ptch1^f/f^* ENS progenitors increases the formation of S100β+ EGCs at the expense of TH+ enteric neurons [60]. Tilting the GLI^A^-*vs*-GLI^R^ balance toward GLI activation in *Wnt1-Cre;Sufu^f/f^* embryos or GLI repression in *Gli3^Δ699/Δ699^* embryos increases or decreases the production of FABP7+ EGCs, respectively [59].
LGI4/ADAM22 Signaling pathway	ENS progenitors and EGCs [80].	Mice deficient in either *Lgi4* or *Adam22* exhibit a similar defect in enteric gliogenesis, characterized by a decreased number of FABP7+ EGCs in vivo and lower GFAP expression in enteric neurosphere assays [80].
Notch Signaling pathway	NCCs [81,82], SCPs [83] and ENS progenitors [63] for multiple DLL/JAG ligands and Notch receptors.	Targeted inhibition of Notch signaling results in a marked decrease of FABP7+ EGCs in *Wnt1-Cre;Rbpsuh^fl/fl^* embryos, which is accompanied by a more modest decrease in the number of TuJ1+ enteric neurons [84]. DLL1 treatment of cultured ENS progenitors is sufficient for promoting the formation of GFAP+ EGCs, while DAPT-mediated inhibition of Notch signaling impairs Hedgehog-induced gliogenesis in the same system [60].
NRG/ERBB Signaling pathway	NCCs [85], SCPs [86], ENS progenitors and EGCs for ERBB3 receptor [87]. Gut mesenchyme for NRG1 (GGF2) ligand [87,88].	S100β staining suggest that both SCPs and EGCs are absent in *erbB3^−^*^/*−*^ embryos [89]. NRG1 (GGF2) treatment of cultured ENS progenitors promote their differentiation in GFAP+ EGCs, this effect being increased by pre-treatment with BMP4 [87].

**Table 5 ijms-24-12475-t005:** Multipotency analyses of mature EGCs in mice.

References	Experimental Condition	Relevant Results
Joseph et al., 2011 [113]	CD49b+ EGCs sorted from the small intestine (muscles and myenteric plexus) of adult WT mice.	Sorted CD49b+ cells express glial markers (GFAP, SOX10, S100β, p75, and Nestin) and can be cultured as self-renewing neurospheres that differentiate in peripherin+ neurons, GFAP+ EGCs and α-SMA+ myofibroblasts.
	BrdU incorporation assays in the small intestine of adult WT mice (and rats) housed in normal conditions or exposed to various potential triggers of neurogenesis (e.g., DSS-induced inflammation, BAC-induced focal aganglionosis).	Basal enteric gliogenesis is detectable under steady-state condition, becoming markedly increased after certain types of injury (up to 90% of S100β+ were also BrdU+ in BAC-ablated regions). No evidence of neurogenesis, with exception of a single rat (out of 85 rodents in total) in which 6.1% of HuC/D+ myenteric neurons did incorporate BrdU in BAC-ablated region.
	Cell lineage tracing in the small intestine of adult *GFAP-Cre;R26R-YFP* or *GFAP-CreERT2;R26R-YFP* mice, exposed to BAC treatment or not.	With the constitutive Cre driver line, 6–7% of HuC/D+ myenteric neurons were also YFP+ in both control and BAC-treated mice. This most likely reflects an early fetal/neonatal contribution from a GFAP+ progenitor, which was no longer detectable when the tamoxifen-inducible Cre driver was activated in adults (<0.1% of HuC/D+ also YFP+ in this case).
Laranjeira et al., 2011 [116]	Cultures of enzymatically dissociated small intestine (muscles and myenteric plexus) from tamoxifen-treated adult *Sox10-iCreERT2;R26R-YFP* or *hGFAP-CreERT2;R26R-YFP* mice.	YFP+ cells generate bipotential SOX10+ PHOX2B+ ASCL1+ ENS progenitors that can be cultured as self-renewing neurospheres, and can be differentiated in GFAP+ EGCs and multiple neuronal subtypes (nNOS+, VIP+, or NPY+).
	Cell lineage tracing studies in the small intestine of adult *Sox10-iCreERT2;R26R-YFP* mice, exposed to BAC treatment or not.	YFP+ HuC/D+ myenteric neurons are not detected following tamoxifen treatment under steady-state conditions but are readily detected upon BAC-mediated ENS ablation.
Belkind-Gerson et al., 2013 [117]	Neurospheres prepared from enzymatically dissociated colon (mucosa and submucosal plexus vs. muscles and myenteric plexus) of *Nestin-GFP* mice.	GFP+ cells co-express glial markers (S100β, GFAP) in vivo, and generate neurospheres containing TuJ1+ neurons and S100β+ EGCs that both co-express GFP in culture.
Belkind-Gerson et al., 2015 [115]	Pseudo cell lineage tracing studies in colon of *Sox2-GFP* and *Nestin*-GFP mice, exposed to DSS treatment or not.	In absence of DSS, GFP expression is virtually undetectable in HuC/D+ myenteric neurons but becomes detectable 48 h after DSS treatment (8% of neurons in *Sox2-GFP* vs. 1.8% in *Nestin-GFP* mice).
	Culture of CD49+ EGCs sorted from small intestine and colon (muscles and myenteric plexus) of adult mice, in absence or presence of a serotonin receptor antagonist	Sorted CD49b+ EGCs generate TuJ1+ neurons, GFAP+ EGCs and TuJ1+ GFAP+ neuroglial cells in culture. The serotonin receptor antagonist increases the proportion of these neuroglial cells at the expense of neurons.
	Transplantation of neurospheres derived from CD49b+ EGCs in explants of aneural embryonic chick hindgut	Transplanted neurospheres generate TuJ1+ neurons and GFAP+ EGCs in both myenteric and submucosal plexus.
Belkind-Gerson et al., 2017 [114]	Cell lineage tracing studies in colon of adult *Sox2-CreERT2:R26R-YFP* and *Plp1-CreERT2:R26R-tdTomato* mice, exposed to DSS treatment or not.	DSS treatment increases the proportion of HuC/D+ myenteric and submucosal neurons co-expressing either of the fluorescent reporters in tamoxifen-induced mice.
	Neurospheres prepared from enzymatically dissociated colon (full thickness) of adult tamoxifen-treated *Plp1-CreERT2;R26R-tdTomato* mice.	tdTomato is expressed in neurons (either TuJ1+, HuC/D+, or PGP9.5+), EGCs (either SOX2+ or S100β+), and neuroglial cells co-expressing neuronal and glial markers.
Guyer et al., 2023 [30]	Neurospheres prepared from enzymatically dissociated small intestine (muscles and myenteric plexus) of adult *Plp1-GFP;Actl6b-Cre;R26R-tdTomato* dual reporter mice.	GFP+ EGCs sorted from neurospheres generate new tdTomato+ neurons in culture.
	Sorted tdTomato-negative cells from small intestine (muscles and myenteric plexus) of adult *Actl6b-Cre;R26R-tdTomato* mice.	Neurospheres derived from sorted tdTomato-negative cells generate new tdTomato+ neurons in culture.

Abbreviations: BAC, benzalkonium chloride; BrdU, bromodeoxyuridine; DSS, dextran sodium sulfate.

## Data Availability

All relevant data are included in the article.

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
