# Peer review of "Harnessing the Power of Enteric Glial Cells’ Plasticity and Multipotency for Advancing Regenerative Medicine"

_ijms, 2023, doi:10.3390/ijms241512475_

Round 1

Reviewer 1 Report

Summary:

In this review, Lefevre et al gave a comprehensive and thought-provoking summary of the important roles of enteric glial cells (EGCs), covering the ontology or source of EGC, functional heterogeneity, plasticity and stemness, and their therapeutic application. The review not only organizes recent literature related to EGCs in a logic flow, but also gives expert opinions on recent studies and trends. I have no other comments and highly support this current form for publication.

Author Response

Thank you for these highly supportive comments! Indeed, our primary intention was to go beyond a simple review of the literature, to instead share our views on what we believe to be key issues to be addressed in the field.

Reviewer 2 Report

In this review manuscript Lefévre et al., summarize the recent knowledge of mammalian enteric glia cells. Greater understanding of the ENS biology, emphasizing the origin, morphology and complex nature of the enteric glia types is a great value to both developmental biology and regenerative field.

I have only minor comments:

1)    a schematic illustration of enteric glia types and their distribution within the gut wall would help the better understanding of this complex system.

2)    detailed summary of non-neuronal origin of enteric neurons (last paragraph of section 2; lines 88-96) is less relevant to this paper.

3)    Please include a short section about the immunological role of EGCs.

4)    I would recommend for the authors to cite this recent paper highly connected to Section 3:

Recent observations of Stavely et al (2023) indicate that, over developmental maturation, ENS progenitors show transcriptomic features more consistent with enteric glia, which are lost in those committed to becoming enteric neurons.

Stavely, R., Hotta, R., Guyer, R. A., Picard, N., Rahman, A. A., Omer, M., Soos, A., Szocs, E., Mueller, J., Goldstein, A. M., & Nagy, N. (2023). A distinct transcriptome characterizes neural crest-derived cells at the migratory wavefront during enteric nervous system development. Development (Cambridge, England), 150(5), dev201090. https://doi.org/10.1242/dev.201090

Please cite the following publications regarding the functional subtypes in humans:

Badizadegan, K., Thomas, A. R., Nagy, N., Ndishabandi, D., Miller, S. A., Alessandrini, A., Belkind-Gerson, J., & Goldstein, A. M. (2014). Presence of intramucosal neuroglial cells in normal and aganglionic human colon. American journal of physiology. Gastrointestinal and liver physiology, 307(10), G1002–G1012. https://doi.org/10.1152/ajpgi.00164.2014

Author Response

1- A schematic illustration of enteric glia types and their distribution within the gut wall would help the better understanding of this complex system.

Response: A new Figure based on our immunofluorescence staining of topo-morphological subtypes of EGCs is now included.

2- Detailed summary of non-neuronal origin of enteric neurons (last paragraph of section 2; lines 88-96) is less relevant to this paper.

Response: We think this section is nonetheless important as the highlighted limitations of these studies help to further emphasize the major contribution of neural crest-derived progenitors.

3- Please include a short section about the immunological role of EGCs.

Response: Thank you for the suggestion. We added a few sentences about this in the Introduction. However, we feel that devoting a complete section to this topic would be somewhat less relevant for the current review which focuses on EGC subtypes, not global functions of EGCs.

4- I would recommend for the authors to cite this recent paper highly connected to Section 3: Recent observations of Stavely et al (2023; https://doi.org/10.1242/dev.201090) indicate that, over developmental maturation, ENS progenitors show transcriptomic features more consistent with enteric glia, which are lost in those committed to becoming enteric neurons.

Response: Thank you for the suggestion. This reference has been added where relevant in Section 3.

5- Please cite the following publications regarding the functional subtypes in humans: Badizadegan, K., et al (2014; https://doi.org/10.1152/ajpgi.00164.2014).

Response: Thank you for the suggestion. However, this article does not contain functional studies and thus cannot be included in our Table 2 that refers to functional subtypes.

Reviewer 3 Report

This manuscript explores the role of enteric glial cells (EGCs) in the enteric nervous system (ENS) and their potential for regenerative medicine. It discusses the origins of the ENS, focusing on neural crest cells as the primary source of EGCs. The manuscript highlights the need for further research on EGC subtypes, their formation, and differentiation mechanisms. It also discusses the plasticity and multipotency of EGCs, their involvement in pathological conditions, and their potential for therapeutic interventions. Overall, the manuscript emphasizes the importance of understanding EGCs for advancing regenerative medicine and improving gastrointestinal health.

The manuscript presents novel content, and the writing is acceptable, but there are areas where the logical flow of the text needs improvement. The Conclusions and Perspectives section of the article requires additional content to be added.

Please find the detailed suggestions in my pdf file.

While the content of the manuscript is innovative, there are instances where the language could be further refined for clarity and precision. Consider revising certain sentences or phrases to ensure they convey the intended meaning accurately.

The overall grammar and syntax are satisfactory, but there are occasional errors or inconsistencies that could be addressed through careful proofreading. Pay attention to verb tenses, subject-verb agreement, and the correct usage of articles and prepositions.

The organization of ideas could be strengthened to enhance the coherence and flow of the text. Consider reordering or restructuring certain paragraphs or sections to create a smoother transition between concepts and arguments.

Although the manuscript demonstrates a solid foundation in English writing, there is room for improvement in terms of word choice and style. Aim for a more concise and precise expression of ideas, avoiding unnecessary repetition or verbosity.

Ensure that the terminology and scientific terms used throughout the manuscript are accurate, consistent, and properly defined. Review the manuscript for any technical jargon that may be unclear to the target audience and provide necessary explanations or definitions.

Author Response

This manuscript explores the role of enteric glial cells (EGCs) in the enteric nervous system (ENS) and their potential for regenerative medicine. It discusses the origins of the ENS, focusing on neural crest cells as the primary source of EGCs. The manuscript highlights the need for further research on EGC subtypes, their formation, and differentiation mechanisms. It also discusses the plasticity and multipotency of EGCs, their involvement in pathological conditions, and their potential for therapeutic interventions. Overall, the manuscript emphasizes the importance of understanding EGCs for advancing regenerative medicine and improving gastrointestinal health. The manuscript presents novel content, and the writing is acceptable, but there are areas where the logical flow of the text needs improvement. The Conclusions and Perspectives section of the article requires additional content to be added. Please find the detailed suggestions below.

Response: the whole manuscript has been thoroughly revised to improve the logical flow and provide more detail for some key aspects, including an expansion of the Conclusion and Perspectives section. 

1- The abstract could be improved if written as follows: The enteric nervous system (ENS), known as the intrinsic nervous system of the gastrointestinal tract, is composed of a diverse array of neuronal and glial cell subtypes. The fascinating questions surrounding the generation of cellular diversity in the ENS have captivated ENS biologists for a considerable time, particularly with recent advancements in cell type-specific transcriptomics at both population and single-cell levels. However, the current focus of research in this field is predominantly restricted to the study of enteric neuron subtypes, while the investigation of enteric glia subtypes significantly lags behind. Despite this, enteric glial cells (EGCs) are increasingly recognized as equally important regulators of numerous bowel functions. Moreover, a subset of postnatal enteric glia exhibits remarkable plasticity and multipotency, distinguishing them as critical entities in the context of advancing regenerative medicine. In this review, we aim to provide an updated overview of the current knowledge on this subject, while also identifying key questions that necessitate future exploration.

Response: Thank you. The abstract has been re-written as per your suggestion.

2- Suggestions to improve the introduction: 1. Provide more context: While the introduction provides a basic overview of the ENS, consider expanding on the importance of the ENS in maintaining gastrointestinal homeostasis and its intricate connection to digestive functions. This will help the reader understand the significance of studying the ENS and its cellular components. 2. Clarify the organization of the ENS: Further elaborate on the structural organization of the myenteric plexus and submucosal plexus, emphasizing their roles in specific gastrointestinal functions. This will enhance the understanding of how the ENS regulates various aspects of gut physiology. 3. Elaborate on bidirectional gut-brain communication: Explain in more detail the mechanisms and importance of bidirectional gut-brain communication mediated by the ENS. This will help establish the relevance of the ENS in influencing both gastrointestinal functions and central nervous system activities. 4. Expand on the diversity of enteric neuron subtypes: Provide additional information about the functional significance of the different enteric neuron subtypes and how they contribute to the complex regulation of gastrointestinal functions. This will highlight the complexity of the ENS and its ability to coordinate various physiological processes. 5. Elaborate on the significance of enteric glial cells (EGCs): Offer more background on the roles of EGCs in supporting and modulating enteric neuron function, as well as their involvement in regulating bowel functions. This will strengthen the understanding of the importance of studying EGCs alongside enteric neurons. 6. Highlight the relevance of plasticity and multipotency: Emphasize the potential implications of the plasticity and multipotency exhibited by certain EGCs in the context of regenerative medicine and cell-based therapies. Discuss how understanding these properties can contribute to the development of innovative therapeutic approaches for gastrointestinal conditions. 7. Provide a clear outline of the review: In the final sentence of the introduction, explicitly state the structure and purpose of the review, including the specific aspects that will be covered and the identified research needs. This will provide a roadmap for the reader and set clear expectations for the rest of the article.

Response: We now provide more context and detail about the ENS, enteric neurons and EGCs, notably including information relative to the central roles played by EGCs in the control of key gastrointestinal functions (immunity and epithelial barrier). We feel that addressing all additional comments would make the introduction far too long and repetitive with the content of other sections.

Other suggestions for improvement:

  1. The ENS is built from multiple progenitors: Consider providing more context and background information about the studies conducted on model organisms to support the consensus that the ENS is primarily derived from neural crest cells (NCCs). Provide a clear explanation of the different subpopulations of NCCs along the anterior-posterior axis and their respective contributions to ENS progenitors. Clarify the timing and migration patterns of vagal and sacral NCCs in mice for a better understanding of their role in generating enteric neurons and glia. Expand on the contribution of non-NCC sources of ENS progenitors, such as the ventral neural tube and gut/pancreas endodermal epithelium, and their timing in relation to vagal NCC colonization.

Response: we feel that all these aspects are already properly covered in Section 2.

  1. Formation and diversification of EGCs: Elaborate on the differentiation process of ENS progenitors into enteric neurons and glia, particularly during gut colonization. Provide more information on the molecular mechanisms involved in the neurogenic-to-gliogenic fate transition and the gene-regulatory network responsible for EGC differentiation. Explore and discuss additional positive regulators and their roles in enteric gliogenesis. Consider discussing the differentiation potential of EGCs into myofibroblasts and the specific circumstances required for this differentiation to occur. Highlight the need for further research to understand the appearance and diversification of different subtypes of EGCs, including the challenges in reconciling different forms of EGC subtypes (morphological/topological, functional, or transcriptional).

Response: we feel that all these aspects are already properly covered in Section 3. To the best of our knowledge, there are no additional positive regulators of enteric gliogenesis that are not already included in the accompanying Table 4.

  1. Plasticity and multipotency of EGCs: Provide more information on the phenotypic plasticity of EGCs under pathological conditions, including changes in molecular composition, structure, and function. Discuss the potential beneficial effects of reactive EGCs and their secretion of GDNF in restoring epithelial barrier integrity and controlling inflammation in gastrointestinal disorders. Expand on the stem cell-like properties of EGCs, including their self-renewal and differentiation into enteric neurons, and the need for further research to understand the regulatory mechanisms and potential therapeutic applications.

Response: we feel that all these aspects are already properly covered in Section 4.

  1. Taking advantage of EGCs' plasticity and multipotency for therapeutic purposes: Provide more examples and discuss specific drugs, nutraceutical products, or therapeutic strategies that have shown promising effects in modulating the pathological effects of reactive EGCs. Discuss the potential of cell transplantation-based therapies for regenerating the damaged/missing ENS, including the challenges and opportunities associated with ENS stem/progenitor cells derived from intrinsic EGCs and extrinsic Schwann cells. Explore the possibility of co-transplanting ENS stem/progenitor cells of different origins for maximizing neuronal and glial diversification. Emphasize the importance of in situ stimulation of tissue-resident ENS stem/progenitor cells and the role of factors like GDNF in promoting neurogenesis and gliogenesis.

Response: we feel that all these aspects are already properly covered in Section 5.

  1. Conclusion and perspectives: Highlight the growing recognition of the importance of EGCs in gastrointestinal functions and their potential implications in neurological disorders. Emphasize the need for further research and advancements in understanding EGC formation, function, and their role in various diseases. Discuss the potential impact of increasing knowledge about EGCs on managing gastrointestinal diseases and neurological disorders.

Response: we have expanded this section to better emphasize the potential impact of EGC research for extra-digestive neurological disorders like Parkinson disease and amyotrophic lateral sclerosis. 

Round 2

Reviewer 3 Report

None.